# Common mouse models of chronic kidney disease are not associated with cachexia
Benjamin Lair[1,3], Marlène Lac [1,3], Lucas Frassin[1], Manon Brunet[2], Marie Buléon[2], Guylène Feuillet[2], Claire Maslo[1], Marie Marquès[1], Laurent Monbrun [1], Virginie Bourlier[1], Emilie Montastier[1], Nathalie Viguerie [1], Geneviève Tavernier[1], Claire Laurens [1] & Cedric Moro [1] ✉

The 5/6 nephrectomy and adenine-induced nephropathy mouse models have been extensively used to study Chronic Kidney Disease (CKD)-related cachexia. One common caveat of these CKD models is the cross-sectional nature of comparisons made versus controls. We here performed a comprehensive longitudinal assessment of body composition and energy metabolism in both models. The most striking finding is that weight loss is largely driven by reduced food intake which promotes rapid loss of lean and fat mass. However, in both models, mice catch up weight and lean mass a few days after the surgery or when they are switched back to standard chow diet. Muscle force and mass are fully recovered and no sign of cachexia is observed. Our data demonstrate that the time-course of kidney failure and weight loss are unrelated in these common CKD models. These data highlight the need to reconsider the relative contribution of direct and indirect mechanisms to muscle wasting observed in CKD.

Chronic kidney disease (CKD) is a global health challenge that affects approximately 10%–15% of people worldwide. CKD represents a state of progressive and irreversible loss of kidney function and its prevalence increases with age, particularly in individuals with hypertension and diabetes[1]. Therapeutic options are currently limited, and CKD progression often leads to the use of peritoneal dialysis, hemodialysis or to kidney transplantation[2]. CKD is frequently associated with severe loss of muscle mass and force that negatively impact the quality of life of patients, leading to higher risks of frailty, co-morbidities and mortality[3]. Several mechanisms have been proposed for a causal link between kidney dysfunction and muscle wasting, such as the protein energy wasting (PEW) syndrome[4,5]. In addition, various co-morbidities, lifestyle factors (diet and physical activity), and more importantly hemodialysis have been suggested to play a key role in CKD-associated PEW. Numerous evidences relating kidney dysfunction to muscle wasting stem from common mouse models of CKD. These models have been extensively used to test potential therapeutic strategies in CKD-associated PEW[6,7]. The 5/6 nephrectomy (Nx) model, which involves surgical resection of kidney mass, is one of the most widely used techniques to successfully induce renal failure in laboratory animals[8–10]. Another model is the adenine diet model of CKD, a nonsurgical option, that was developed to induce renal dysfunction in rodents[11]. Kidney disease with adenine feeding stems from the formation of 2,8-dihydroxyadenine, an adenine metabolite

that crystalizes within renal tubules and causes injury, inflammation, tubular atrophy, and fibrosis of the renal parenchyma[12]. Nx and adenine models seem to produce equivalent levels of nephropathy and muscle atrophy[11]. In experimental mouse studies investigating potential mechanisms of CKD-related cachexia, one caveat is that phenotypes and molecular mechanisms at the tissue level are investigated in a cross-sectional manner within usually 3–6 weeks of kidney surgery or diet supplementation[3,11–13]. For instance, previous studies failed to observe significant changes in body weight after inducing CKD when pair-feeding control sham animals[14]. We therefore hypothesized that changes in body composition and energy metabolism could be unrelated to kidney dysfunction. To test this idea, we here comprehensively investigated longitudinal changes in body composition and energy metabolism in two common CKD mouse models. Our data thus unambiguously show that the time-course of kidney failure and weight loss are unrelated, and that no sign of muscle wasting is observed in common CKD mouse models.

## Results

### Longitudinal changes in body weight and composition in the Nx model

We first performed a 9 week longitudinal follow-up of changes in body weight and composition in sham-operated and Nx mice (Fig. 1a). 5/6

[1]Team MetaDiab, Institute of Metabolic and Cardiovascular Diseases, INSERM/Paul Sabatier University UMR1297, Toulouse, France. [2]Team Renal Fibrosis and Chronic Kidney Diseases, Institute of Metabolic and Cardiovascular Diseases, INSERM/Paul Sabatier University UMR1297, Toulouse, France. [3]These authors contributed equally: Benjamin Lair, Marlène Lac. ✉e-mail: cedric.moro@inserm.fr

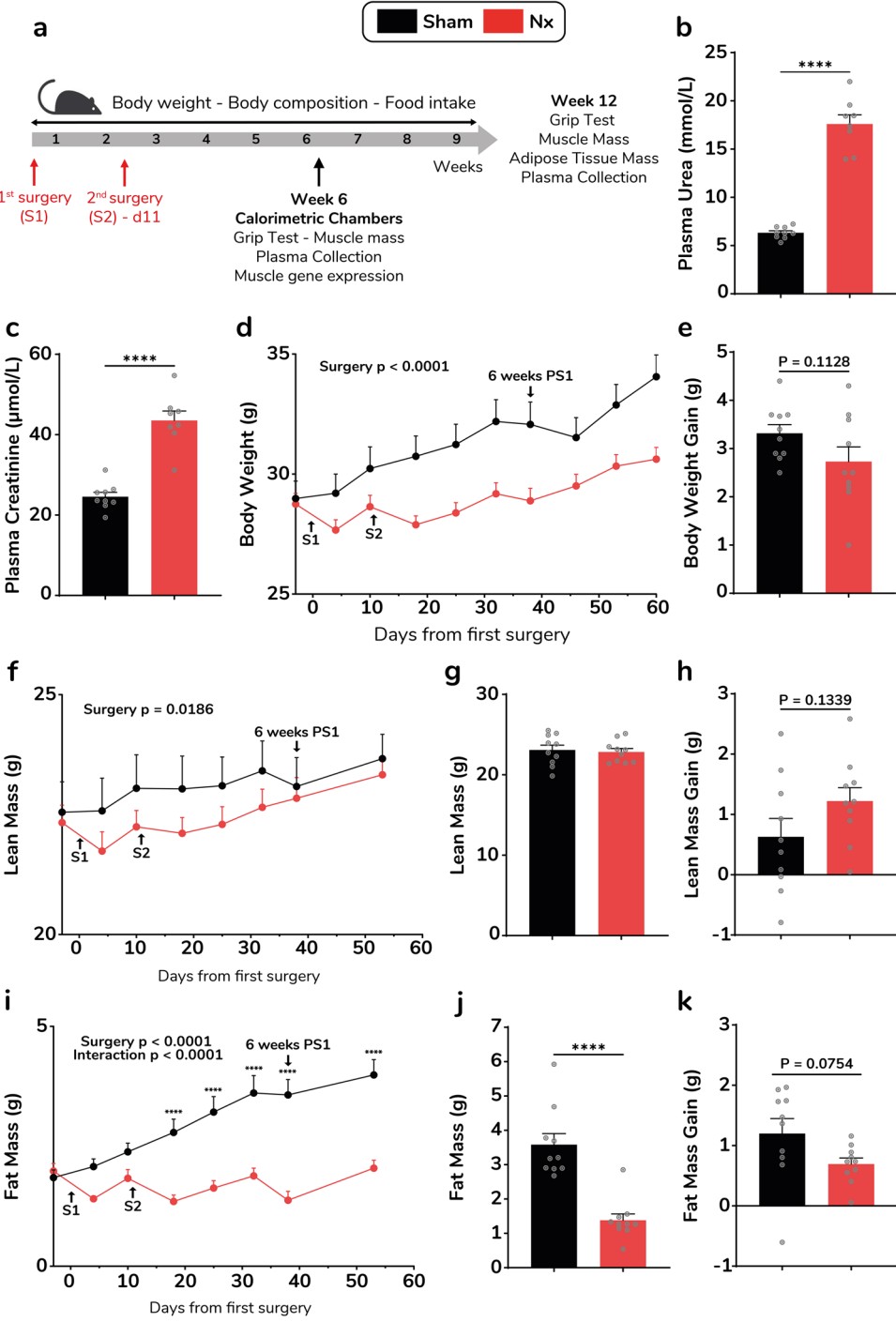

**Fig. 1 | Time-course of body weight and composition changes in the Nx model.** Experimental design of the 5/6-nephrectomy model (Nx) (**a**). Plasma urea (**b**) and plasma creatinine (**c**) in sham (black bars) versus Nx mice (red bars) 6 weeks after the first surgery (PS1). Longitudinal changes in body weight over 9 weeks (**d**) and body weight gain over the recovery period (weeks 3–9) (**e**) in sham (black bars/lines) versus Nx mice (red bars/lines). Longitudinal changes in lean mass (**f**), lean mass at 6 weeks after the first surgery (**g**) and lean mass gain over the recovery period (weeks 3–8) (**h**). Longitudinal changes in fat mass (**i**), fat mass at 6 weeks after the first surgery (**j**) and fat mass gain over the recovery period (weeks 3–8) (**k**) in sham versus Nx mice ($n = 10$ mice per group). Data are expressed as means ± s.e.m. Statistical significance was calculated by unpaired $t$-tests and two-way ANOVA accordingly. ****$p < 0.0001$.

nephrectomy was achieved through two consecutive surgical steps over 11 days. We first used adult 12 week old C56Bl/6 J male mice. CKD was confirmed by a 2.8-fold increase of plasma urea ($p < 0.0001$) (Fig. 1b) and 1.8-fold increase of plasma creatinine levels ($p < 0.0001$) (Fig. 1c) 6 weeks post first surgery (PS1). Elevated levels of plasma urea (Supplementary Fig. 1a) and creatinine (Supplementary Fig. 1b) were still present 12 weeks after the first surgery (PS1). When examining the time-course of body weight, both surgeries induced weight loss (Fig. 1d). However, Nx mice started to catch-up weight within one week after the second surgery. Body weight gain 9 weeks PS1 was not statistically different in sham and Nx mice (Fig. 1e). Cachexia primarily defines a loss of lean mass[15]. We here observed a negative effect of kidney surgeries on total lean mass (Fig. 1f). However, Nx mice quickly catch-up lean mass one week after the second surgery to reach

a comparable final total lean mass (Fig. 1g). Thus, lean mass gain during the recovery period (weeks 3–9) was comparable between Nx and sham mice (Fig. 1h). In contrast to lean mass, fat mass remained lower up to 9 weeks PS1 (Fig. 1i-k). This difference was even more pronounced 12 weeks PS1, as total body weight of Nx mice remained lower (Supplementary Fig. 1c) due to reduced fat mass as evidenced by lower inguinal and epididymal fat pad weight (Supplementary Fig. 1d).

## Longitudinal changes in whole-body energy metabolism in the Nx model

To further understand changes in body weight and composition in the Nx model, we monitored food intake weekly and investigated whole-body energy metabolism in calorimetric chambers 6 weeks PS1. In agreement

**Fig. 2 | Time-course of food intake and whole-body energy metabolism in the Nx model.** Longitudinal monitoring of food intake over 9 weeks after the first surgery in sham (black bars) and Nx (red bars) mice (**a**). Water intake (**b**), cumulative food intake (**c**) and average 24 h food intake (**d**) measured over 24 h in metabolic cages 6 weeks PS1. Changes in 24 h energy expenditure (**e**) and average energy expenditure (**f**) over 24 h in sham (black bars/lines) versus Nx (red bars/lines) mice ($n = 6$–8 mice per group). Data are expressed as means ± s.e.m. Statistical significance was calculated by unpaired $t$-tests and two-way ANOVA accordingly. ***$p < 0.0001$.

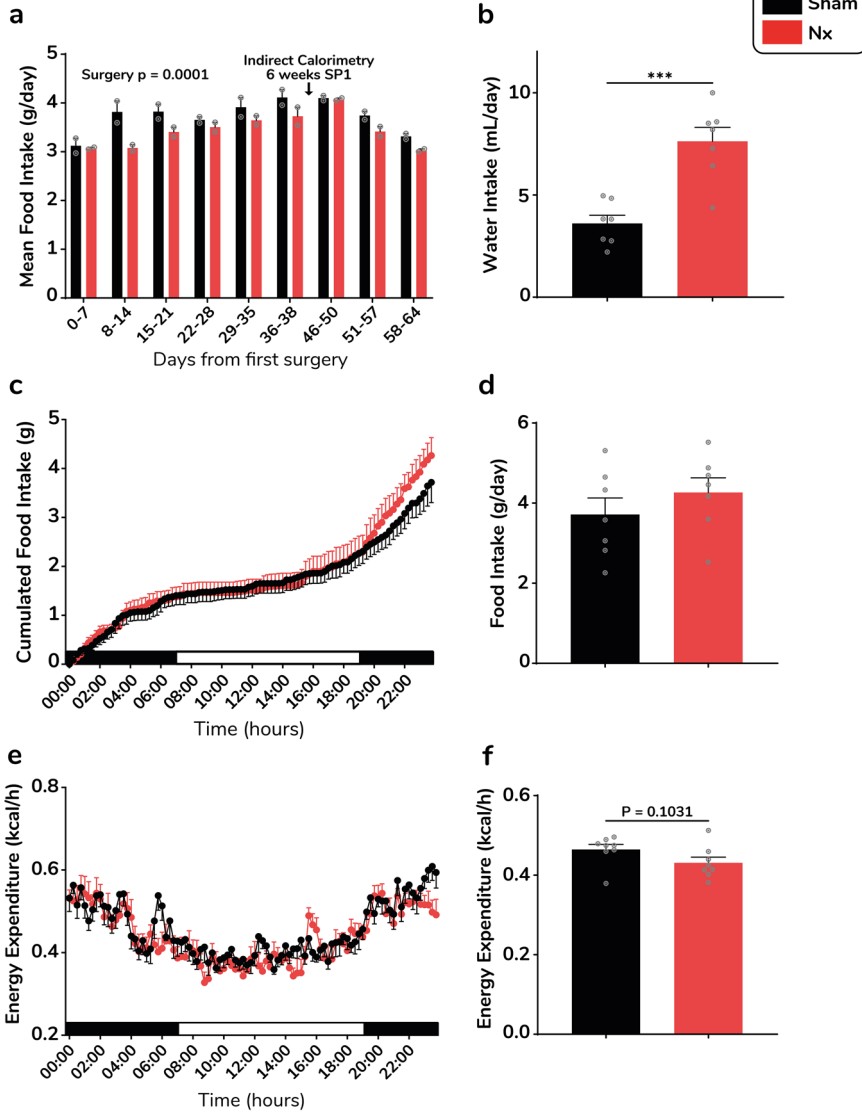

with body weight and composition data, we observed a 20% lower food intake following the second surgery at days 8–14 (3.817 vs 3.078 g/day in sham vs Nx mice) (Fig. 2a). Food intake appeared overall lower from the first surgery to the end of the protocol in Nx vs sham mice ($-6.5\%$, $p < 0.0001$) (Fig. 2a). In agreement with previous observations, 24 h water intake was nearly 2-fold higher in Nx mice 6 weeks PS1 (Fig. 2b)[16]. The time-course of food intake (Fig. 2c) and cumulative food intake (Fig. 2d) measured in metabolic chambers over 24 h were identical between sham and Nx mice, 6 weeks PS1. Similar observations were made for 24 h whole-body energy expenditure measured 6 weeks PS1 (Fig. 2e-f). Consistent with lower total fat mass, the pattern of RER over 24 h was significantly higher in Nx (Supplementary Fig. 1e), but the overall average 24 h RER remained similar (Supplementary Fig. 1f). This was independent of changes in locomotor activity (Supplementary Fig. 1g). Despite lower fat mass, no significant change in plasma triglyceride (TG) (Supplementary Fig. 1h) and free fatty acid (FFA) (Supplementary Fig. 1i) were observed in Nx mice.

## Skeletal muscles from Nx mice do not display features of cachexia

Since cachexia primarily defines a loss of lean and muscle mass associated with weakness[17], we next investigated changes in muscle mass and strength 6 and 12 weeks PS1 in sham and Nx mice (Fig. 3). Grip strength was fully preserved 6 weeks PS1 (Fig. 3a). Individual mass of various hindlimb

muscles was strictly comparable in Nx compared to sham (Fig. 3b). In line with this result, muscle histology analysis revealed that *gastrocnemius* average muscle fiber cross-sectional area (CSA) was preserved in Nx mice 6 weeks PS1 (Fig. 3c–d). In agreement with preserved muscle function and mass, no change in relative expression of atrogenes (Supplementary Fig. 2a) and proinflammatory (Supplementary Fig. 2b) genes were observed in *gastrocnemius* muscle of Nx versus sham mice 6 weeks PS1. Importantly, grip strength (Fig. 3e), mass of various hindlimb muscles (Fig. 3f) and gastrocnemius average fiber CSA (Fig. 3g, h), remained fully preserved 12 weeks PS1. No significant change in atrogenes and proinflammatory gene expression were observed in *soleus* muscle 6 weeks PS1 (Supplementary Fig. 2c, d). Plasma TNFα and CCL2 levels were significantly elevated 6 weeks PS1 in Nx mice compared to sham mice (Supplementary Fig. 2e), but remained in a low physiological range as observed in obesity-associated low-grade systemic inflammation[18]. No significant changes in plasma IL6 and IL1β were noted (Supplementary Fig. 2e).

## Age at the time of first surgery does not impact the outcome in the 5/6 nephrectomy model

In order to investigate whether the age of the animal at the time of the surgical procedure may determine muscle health trajectory, we performed 5/6 nephrectomy on 6 weeks and 12 weeks-old mice simultaneously. As observed in the first cohorts, though 5/6 nephrectomy significantly altered

**Fig. 3 | Muscle phenotype in the Nx model.** Grip strength (**a**) and total mass of *tibialis anterior, gastrocnemius* and *quadriceps* muscles (**b**) in sham (black bars) versus Nx (red bars) mice 6 weeks PS1 (*n* = 8–10 mice per group). Representative images (**c**) and muscle fiber cross-sectional area (**d**) of laminin-stained gastrocnemius cryosections from sham and Nx mice euthanized 6 weeks PS1 (*n* = 8 mice per group). Grip strength (**e**) and total mass of *tibialis anterior, gastrocnemius* and *quadriceps* muscles (**f**) in sham versus Nx mice 12 weeks PS1 (*n* = 10 mice per group). Representative images (**g**) and muscle fiber cross-sectional area (**h**) of laminin-stained gastrocnemius cryosections from sham and Nx mice euthanized 12 PS1 (*n* = 7-8 mice per group). Data are expressed as means ± s.e.m. Statistical significance was calculated by unpaired *t*-tests and two-way ANOVA accordingly.

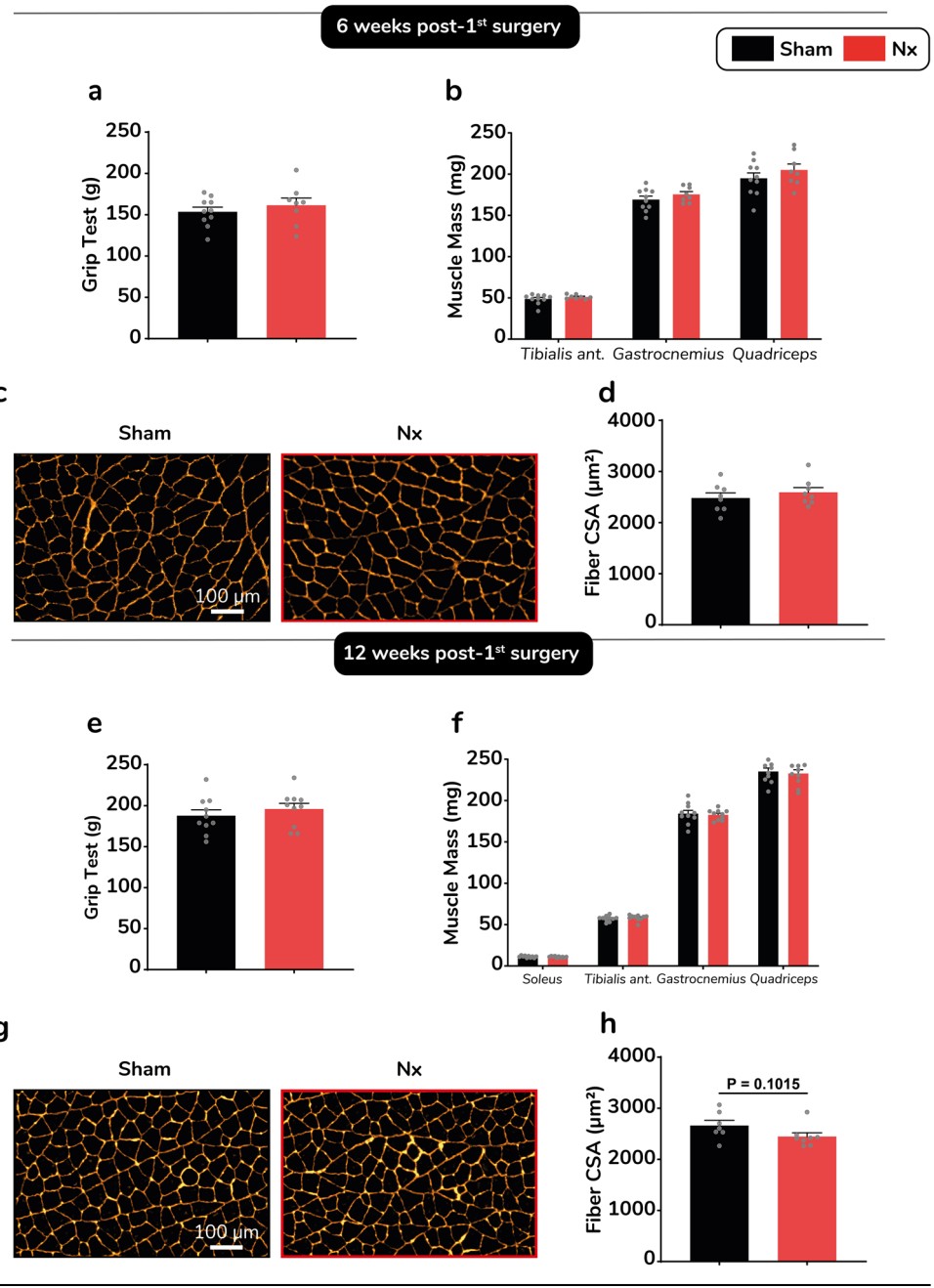

body weight (Fig. 4a), fat mass (-60%, *p* < 0.0001) (Fig. 4b), lean mass (Fig. 4c) as well as individual muscle mass (Fig. 4d) and grip strength (Fig. 4e) remained unchanged 6 weeks PS1 in mice in which nephrectomy procedure was initiated at 12 weeks of age. This occurred despite a drastically decreased glomerular filtration rate (GFR) in Nx mice (-72%, *p* < 0.0001) (Fig. 4f). The same observations were made in mice in which nephrectomy procedure was initiated at 6 weeks of age. Nx significantly lowered body weight (Fig. 4g) and fat mass (Fig. 4h), while lean mass (Fig. 4i) remained strictly unchanged between Nx and sham animals. Consistently, mass of various hindlimb muscles (Fig. 4j) and grip strength (Fig. 4k) were preserved 6 weeks PS1. As previously, GFR was markedly decreased in Nx mice (-60%, *p* < 0.0001) (Fig. 4l).

## Longitudinal changes in body weight, energy metabolism and muscle health in the adenine diet-induced nephropathy model

Another common nonsurgical CKD model is the adenine diet-mediated nephropathy model. Adult male mice were fed a 0.2% enriched adenine diet for 3 non-consecutive weeks (11 days + 1 week) and then switched back to control diet for the rest of the protocol (Fig. 5a). Kidney histology revealed that even 6 weeks after removing adenine diet, mice still displayed severe fibrosis as revealed by increased positive aniline blue surface in kidney sections (0.32% vs 5.3%, *p* = 0.035) (Fig. 5b, c). Adenine-induced kidney dysfunction was confirmed by a significant increase of plasma urea (4.1 fold, *p* < 0.0001) 3 weeks (day 21) after introducing the adenine diet (Fig. 5d). Plasma urea remained elevated in the first days following adenine removal (day 25). Kidney dysfunction at this time point was further confirmed by elevated plasma creatinine levels (1.94 fold, *p* < 0.0001) (Fig. 5e). However, plasma urea dropped remarkably within 2 weeks and remained 1.6 fold elevated in adenine compared to control mice (*p* = 0.054) (Fig. 5d) and water intake was higher compared to control mice as measured over 24 h in calorimetric chambers (1.87 fold, *p* = 0.0006) (Supplementary Fig. 3e). Within 2 weeks after introducing the adenine diet, mice had lost ~25% of their initial body weight (Fig. 5f). For ethical reasons, they were placed back under standard chow diet and significantly regained weight within 3 days,

**Fig. 4 | Time-course of body weight, body composition changes and muscle phenotype as a function of age at surgery.** Longitudinal changes in body weight (**a**), fat mass (**b**) and lean mass (**c**) over 6 weeks PS1 performed in 12 week-old sham (black bars/lines) and Nx (red bars/lines) mice (*n* = 8–10 mice per group). Muscle mass (**d**), grip strength (**e**) and glomerular filtration rate (**f**) 6 weeks PS1 performed in 12 week-old mice (*n* = 8–10 mice per group). Longitudinal changes in body weight (**g**), fat mass (**h**) and lean mass (**i**) over 6 weeks PS1 performed in 6 week-old mice (*n* = 10 mice per group). Muscle mass (**j**), grip strength (**k**) and glomerular filtration rate (**l**) 6 weeks PS1 performed in 6 week-old mice (*n* = 10 mice per group). Data are expressed as means ± s.e.m. Statistical significance was calculated by unpaired *t*-tests and two-way ANOVA accordingly. ***$p < 0.0001$, ****$p < 0.0001$.

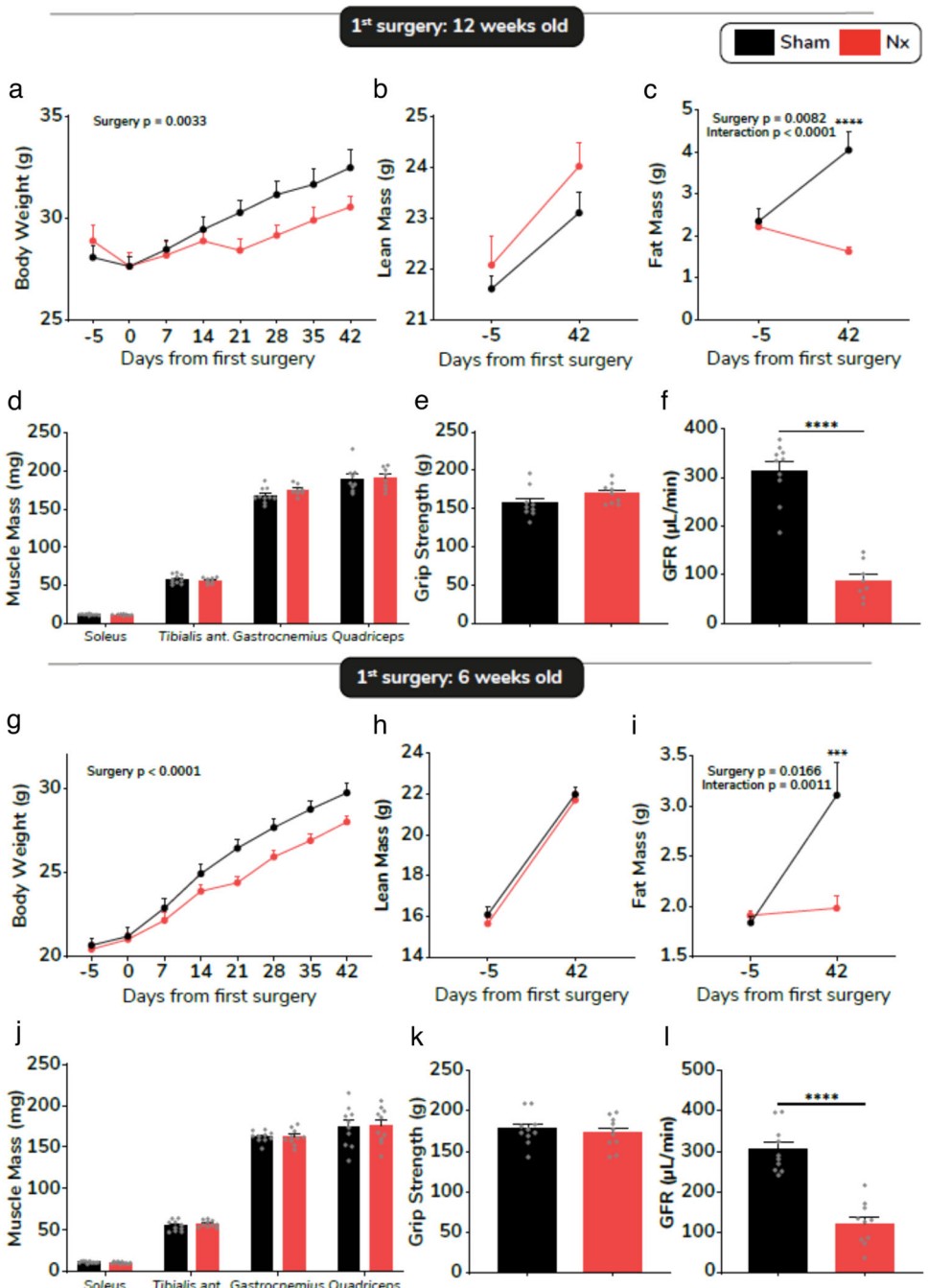

before being switched back to the adenine diet for one extra week. During this extra-week under adenine diet, mice lost ~5 g (Fig. 5f). These changes in body weight were largely reflected by food deprivation that reached 42% during week 1, 68% during week 2 and more than 70% during week 3 of the adenine diet (Fig. 5g). Food intake was completely recovered and tended to increase when mice were switched back to standard chow diet. Thus, mice displayed impressive weight regain up to 6 weeks after discontinuing the adenine diet (Fig. 5f and Supplementary Fig. 3h). However, due to the body weight gap produced by the major energy deficit under the adenine diet, mice of the adenine group displayed a lower body weight compared to control mice at the end of the 9 weeks follow-up (Fig. 5f Supplementary Fig. 3g). Interestingly, mice displayed rapid weight gain between day 21 and day 25 (Fig. 5f). 3 weeks after returning to standard diet, 24 h food intake, energy expenditure and locomotor activity (Supplementary Fig. 3b, d, and f) measured in metabolic chambers were similar in adenine and control diet

mice. Of note, cumulative food intake and energy expenditure measured over a 24 h period (Supplementary Fig. 3a, b) recorded over 24 h in metabolic chambers was not significantly different in adenine and control diet mice. Consistent with changes in body weight, the adenine diet induced a major loss of lean body mass (−22%, *p* < 0.0001) (Supplementary Fig. 4a) and fat mass (−67%, *p* < 0.0001) (Supplementary Fig. 4d) within the first 2 weeks. 3 weeks after switching back to standard diet, mice recovered a substantial amount of their initial lean and fat mass, although they both remained significantly lower compared to control group. At the end of the follow-up, fat mass was still lower in adenine mice as reflected by inguinal and epididymal fat pad weight (Supplementary Fig. 4i). Importantly, individual mass of hindlimb muscles (Fig. 5h) and grip strength (Fig. 5i) were similar between adenine and control diet mice, demonstrating that muscle mass and force were fully recovered a few weeks after discontinuing the adenine diet.

**Fig. 5 | Time-course of body weight changes, food intake and muscle phenotype in the adenine model.** Experimental design of the adenine diet model (**a**). Representative images of kidney sections from control and adenine mice with Masson trichrome staining, 9 weeks after the adenine diet was first introduced (**b**). Quantification of aniline blue positive surface in kidney sections from control (black bars) and adenine (orange bars) mice, 9 weeks after the adenine diet was first introduced (n = 3–5 mice per group) (**c**). Time-course of plasma urea (**d**) and creatinine (**e**) in control (black lines/bars) and adenine (orange lines/bars) mice (n = 6–10 mice per group). Longitudinal changes in body weight (n = 10 mice per group) (**f**). Longitudinal monitoring of food intake (n = 15 mice per group) (**g**). Grip strength (**h**) and total mass of *soleus, tibialis anterior, gastrocnemius* and *quadriceps* muscles (**i**) in adenine vs control mice 9 weeks after first introducing the adenine diet (n = 10 mice per group). Data are expressed as means ± s.e.m. Statistical significance was calculated by unpaired *t*-tests and two-way ANOVA accordingly. *p < 0.05, **p < 0,01, ***p < 0.0001, ****p < 0.0001.

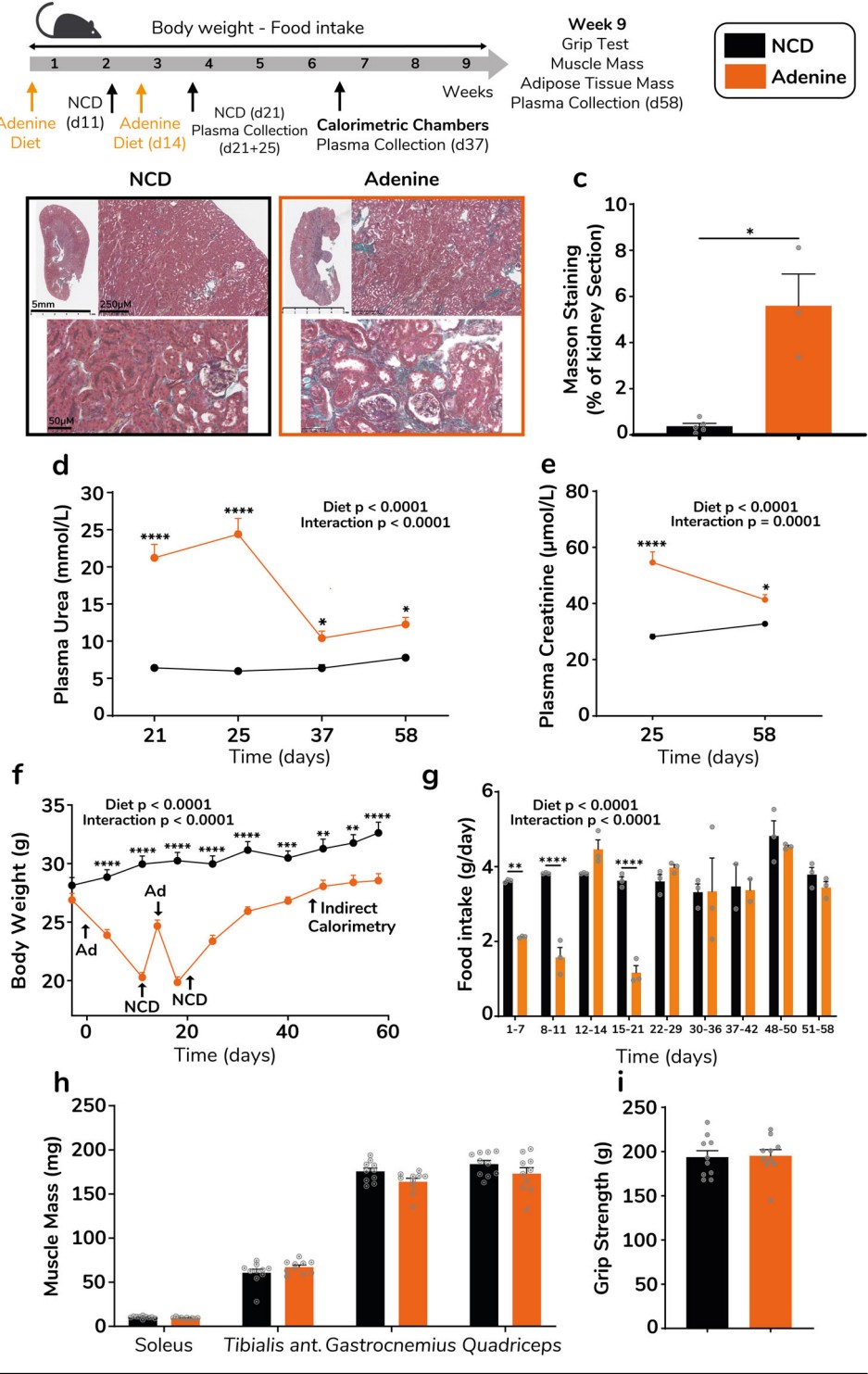

## Long term follow-up in the adenine diet-induced nephropathy model

We were able to monitor adenine-fed and control mice over a longer period of time. Kidney histology analyses revealed that 25 weeks after removal of the adenine supplementation, no improvement in fibrosis was observed (0.36% vs 5.73% p = 0.0079) (Fig. 6a, b). Accordingly, GFR was drastically reduced in adenine fed mice (-74%, p = 0.0079) (Fig. 6c), though plasma urea and creatinine were only modestly elevated compared to control mice (Fig. 6d, e). 25 weeks after switching mice back to standard chow diet, body weight was significantly reduced compared to control mice (Fig. 6f), but this was only attributable to a smaller fat mass (Fig. 6h) while lean mass was

similar between groups (Fig. 6g). In line with body composition data, individual muscle mass and grip strength were once again not altered in adenine-fed mice, even after more than 6 months of kidney dysfunction (Fig. 6i, j).

## Discussion

Several investigators took advantage of 5/6 nephrectomy and adenine-induced nephropathy as preclinical mouse models of CKD. These models have been extensively used to decipher mechanisms of CKD-associated muscle wasting and explore potential therapeutic avenues. A recent study comparing both CKD models suggested that they produce equivalent levels

**Fig. 6 | Long-term follow-up of kidney function, body weight, body composition and muscle phenotype in the adenine model.** Representative images of kidney sections from control and adenine mice with Masson trichrome staining, 25 weeks after discontinuation of the adenine supplementation (**a**). Quantification of aniline blue positive surface in kidney sections from control (black bars) and adenine (orange bars) mice ($n = 5$ mice per group) (**b**). Glomerular filtration rate (**c**) plasma urea (**d**) and plasma creatinine (**e**) in adenine and control mice, 25 weeks after discontinuation of the adenine supplementation ($n = 5$ mice per group). Body weight (**f**), lean mass (**g**) and fat mass (**h**) in adenine and control mice, 25 weeks after discontinuation of the adenine supplementation ($n = 5$ mice per group). Total mass of *soleus*, *tibialis anterior*, *gastrocnemius* and *quadriceps* muscles (**i**) and grip strength (**j**) in adenine vs control mice 25 weeks after discontinuation of the adenine supplementation ($n = 5$ mice per group). Data are expressed as means ± s.e.m. Statistical significance was calculated by Mann-Whitney test and two-way ANOVA accordingly. ** $p < 0.01$.

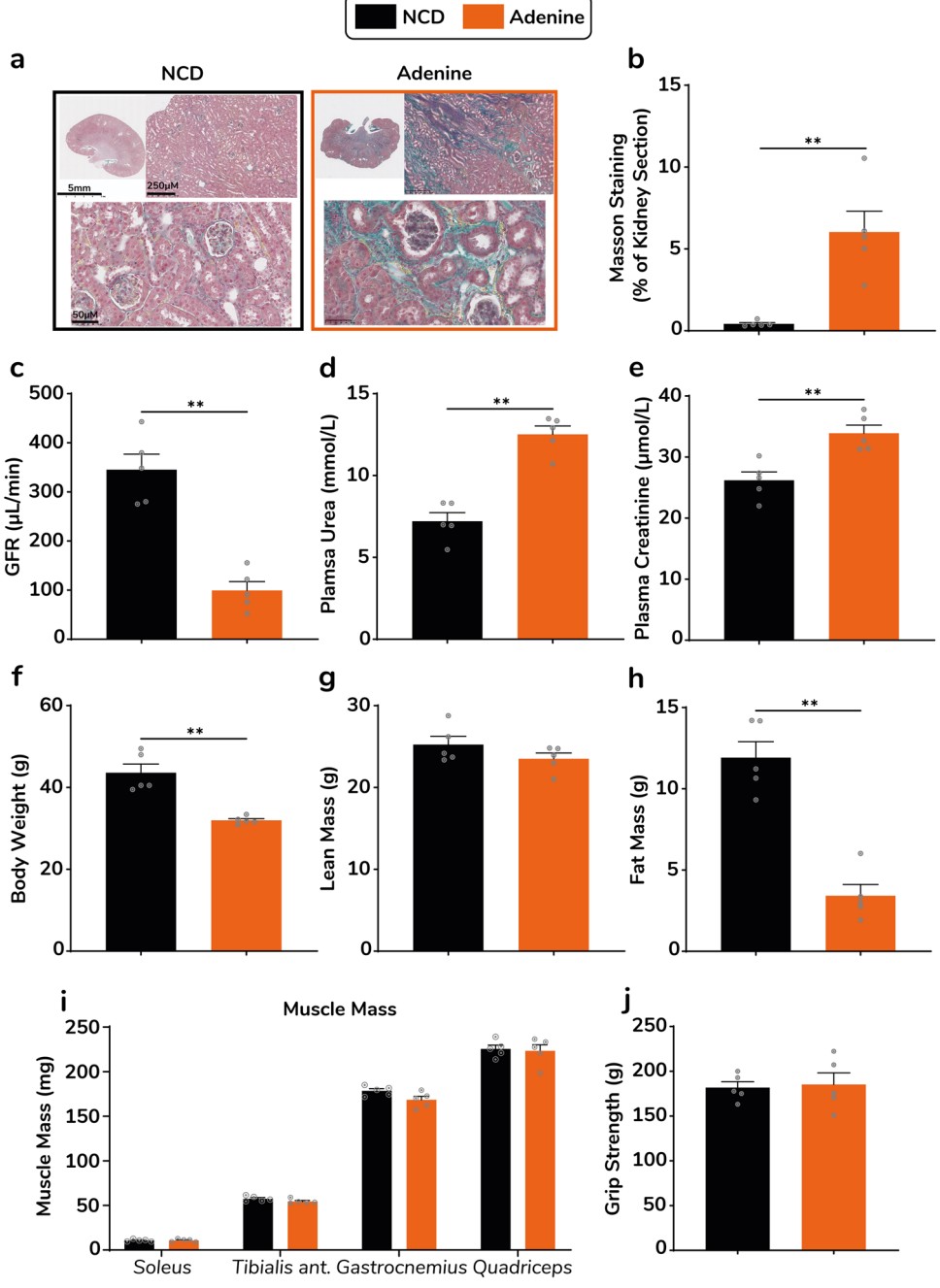

of kidney dysfunction and increased levels of uremic toxins such as kynurenine, kynurenic acid, and indoxyl sulfate[11]. In the current study, overt CKD was reflected by elevated plasma urea, creatinine and water intake in all models. Furthermore, we demonstrate that GFR, a gold standard measure of kidney function, was severely impaired in Nx mice, as well as in adenine-fed mice even several months (up to 6 months) after reintroducing a normal chow diet. Kidney failure was also confirmed by marked alterations of kidney parenchyma and high degree of fibrosis. This is in agreement with previous studies where GFR was shown to be reduced by ~60–70% concomitantly with elevated blood urea nitrogen in both Nx and adenine models[10,11,19,20]. Water intake has also been reported to be significantly greater in Nx mice[16]. However, none of these CKD mouse models trigger muscle wasting as muscle force and mass remain fully preserved. At the molecular level, no major change in atrogenes *Fbxo32* (Atrogin-1) and *Trim63* (Murf-1) expression was noticed in *gastrocnemius* and *soleus*

muscles, two key canonical genes encoding E3-ubiquitin ligases, involved in muscle protein catabolism[21]. Up to 6 months of kidney failure in the adenine model also fail to induce muscle wasting despite lower body weight. Thus, while lean mass is fully preserved, fat mass is severely blunted by the energy deficit (reduced food intake) induced by CKD.

Indeed, the most striking finding is that both Nx and adenine diet were accompanied by a sustained decrease of food intake relative to respective controls. The acute suppression of food intake in the Nx model after the second surgery was ~20%. In contrast the average food intake was dramatically reduced (60–70%) in adenine diet-fed mice mainly due to their reluctance to consume adenine-enriched food. We estimated daily food consumption in the adenine diet to be in average 1.5 g/day which corresponds to 2-3 times less than what control mice usually consume (3.5–4 g/ day). Researchers have tried to increase the palatability of adenine-enriched diets by supplementing chow diet with 20% casein, however these modified

diets fail to preserve food intake and usually lead to major weight loss[11,22]. Net protein loss, reflected by major loss of lean mass in the adenine model is equivalent to long-term fasting in mice[23–25]. Previous studies have shown that 24 h fasting induces ~15% weight loss and ~12% lean mass loss at standard housing temperature[23]. The 24 h fasting model has also been widely used in the field of myology to induce significant muscle atrophy[24]. Therefore, both models are characterized by a transient major energy deficit, even more drastic in the adenine diet model. Mice then regain weight and lean mass continuously, once they have recovered from the nephrectomy procedure or are switched back to standard diet, to reach a similar muscle mass and force after few weeks and up to 6 months of renal failure. In contrast with other studies, we here found no change in whole-body energy expenditure when considered as per mouse[26]. Moreover, upon adenine-diet removal, mice display an impressive body weight regain in a time window where plasma urea remains markedly elevated. Altogether, these findings highlight that there is no direct effect of CKD on muscle wasting in these common mouse models.

Of note, the calculated energy deficit during the recovery period after Nx surgery (week 3–9) was in average of 6.5%. Cumulated food intake from week 3–9 was lower by 28 Kcal in Nx mice which could explain a ~4 g body weight difference at the end of the follow-up, which is even higher than the difference in body weight gain during the same period between Nx and sham. In agreement with our finding, other studies also reported lower food intake in Nx mice that was not sufficient to trigger significant weight loss[14]. More strikingly, when sham mice are pair-fed to Nx mice, no significant change in body weight is observed. This strongly support the idea that energy deprivation induced by reduced food intake is the main trigger of weight loss in these common CKD mouse models. One limitation of this study is that we did not monitor food intake over a longer period of time, especially for the long-term adenine model. We cannot determine if chronically reduced food intake was the driver of decreased fat mass gain, although it seems to be the most plausible explanation.

Lastly, we sought to investigate whether age of mice at the time of nephrectomy in the 5/6 Nx model may explain discrepancies between our results and previously published data. In most studies, CKD is induced in rather young mice of 6–8 weeks of age[11,27–29]. This choice might be motivated by the reported high mortality rate following nephrectomy[11]. However, various experimental studies reported elevated serum phosphate, calcium, fibroblast growth factor-23 (FGF23) and parathyroid hormone (PTH) in CKD mouse models[11], thus altering cortical bone mineral density of the femurs[29]. This could lead to altered musculoskeletal growth in younger mice. We here performed surgery on a total of 40 mice, and only 4 died during the follow-up, which gives an overall mortality rate of 10%, despite severely impaired kidney function with a 3-fold reduction of GFR. Mortality was almost exclusively consecutive to the surgical procedure as it occurred within hours following kidney resection. Our data suggest that 5/6 nephrectomy can be safely performed in adult mice. Yet, when surgeries were performed on 6 weeks old mice, no impact on lean mass gain or final muscle mass and strength was observed, ruling out that impaired growth can drive differences between our results and previous reports. Therefore, it is likely that other confounding factors, such as technical skills to perform the surgical procedure, housing conditions, sanitary status of animal care facility or chow diet composition could explain such discrepancies in mortality rate and muscle mass trajectories. However, CKD did not directly impair muscle mass growth or preservation in any of the models used here. Collectively, our data demonstrate a lack of association between kidney function and muscle mass/force in 5/6 nephrectomy and adenine-induced nephropathy CKD mouse models.

Our findings may also be relevant to understand the pathophysiological link between CKD and muscle wasting in humans. In patients with CKD, the severity of kidney dysfunction correlates with the magnitude of muscle mass and strength loss[30,31]. Hence, hemodialysis is known to deplete blood amino-acids, disrupt appetite and interfere with muscle protein metabolism in favor of catabolism[32,33]. In addition, CKD often coexists with other comorbidities such as metabolic diseases (obesity and diabetes) and hypertension[34–36]. These are also independent risk factors for muscle wasting[21,37]. Finally, when accounting for potential confounding factors in CKD, evidence for a direct link between CKD and cachexia are rather scarce in humans. We came across at least one study demonstrating a decrease of muscle protein synthesis in CKD pre-dialysis patients[38]. However, this conclusion is dampened by the fact that groups were not matched for fat mass and that CKD patients did not exhibit muscle atrophy as indicated by similar fat free mass.

In summary, our data indicate that caution should be taken when interpreting changes in body composition, muscle mass and function in preclinical CKD mouse models. We here demonstrate that changes in body weight and composition are largely mediated by energy deficit and unrelated to kidney dysfunction. Thus, mice can fully recover their initial lean body mass and no obvious signs of cachexia or muscle wasting are observed. Although 5/6 nephrectomy and adenine-induced nephropathy may be suitable models to induce CKD in mice, they do not seem appropriate to investigate mechanisms and therapeutic strategies of CKD-related cachexia. Though mice are not just small humans, our results highlight the need to reconsider by which mechanisms CKD may directly affect muscle mass and function. This will be necessary to guide future clinical practice and therapeutic strategies.

## Methods

### Animals

C57BL/6 J male mice were used. Mice were housed in a pathogen-free barrier facility (12 h light/dark cycle) with *ad libitum* access to water and food in standard animal care facility rooms at 21 °C (RT). Three distinct cohorts of mice were used for the Nx model and one for the adenine model. In a first cohort, 12 weeks-old mice were randomly assigned to either sham or nephrectomy group and mice were killed at 6 weeks after first surgery (PS1). In a second cohort, 12 weeks-old mice were randomly assigned to either sham or nephrectomy group and mice were killed at 12 weeks after first surgery (PS1). In the last cohort, surgery was performed simultaneously in mice at either 6 or 12 weeks of age and mice were killed at 6 weeks PS1. Body weight and, for some of the cohorts, food intake, were measured weekly during the entire protocol. Food intake was measured weekly as the total food consumed per cage, as the difference in grams between total food and remaining food in each cage, divided by the number of mice per cage during the entire follow-up. For the Nx model, body composition was also measured weekly by nuclear magnetic resonance using a Minispec (Bruker). Along both protocols, sub-mandibular blood collection was performed and plasma was isolated by centrifugation (3000RPM, 15mns, 4 °C) and kept at -80 °C until subsequent analyses. At the desired time point, mice were killed and blood was collected into EDTA tubes. Muscles and adipose tissues were rapidly excised, weighted and snap frozen in liquid nitrogen before being stored at -80 °C. All experimental procedures were approved by our institutional animal care and use committee CREFRE CEEA122 (protocol# 2016122311033178) and performed according to INSERM guidelines for the care and use of laboratory animals.

### 5/6 nephrectomy

Three independent cohorts of C56Bl/6 J male mice were used. A slightly modified 5/6 Nx model was used to surgically induce renal dysfunction in mice[16]. This consisted of two surgical steps: (1) initial resection of the upper and lower pole of the left kidney during the first surgery and (2) right kidney resection 1 week after the first surgery. All operating procedures were performed under aseptic techniques. Mice were anesthetized with 2–2.2% isoflurane mixture (3% during the induction phase) and placed on a warm pad (SpaceGel, Braintree Scientific) during the surgery. In the supine position, a midline abdominal incision was made, and the left kidney was exposed. The surgeon performed careful separation of perirenal fat, connective tissue, the adrenal gland, and the ureter with blunt forceps. Upper and lower poles were resected, and hemostatic pads were applied immediately to limit bleeding. After checking that there is no bleeding, the abdominal muscle layer and skin were closed with 5–0 sterile non-

absorbable suture using a simple continuous technique. Subsequently, the contralateral right renal Nx was performed 7 days after the first operation. The Nx was performed by reopening the previous midline laparotomy incision, and the right kidney was isolated. Care was taken to preserve the adrenal gland, after which the renal pedicle with the artery, vein, and ureter was ligated. Finally, the right kidney was extirpated by transecting the vessels and ureter immediately distal to the ligature. The abdominal muscle layer and skin were closed with 5–0 sterile non-absorbable suture using a simple continuous technique. Following each surgery, mice were treated with buprenorphine (100 mg/kg body weight). The sham surgery group also underwent two-step surgeries but without pole resection of the left kidney or right kidney resection.

### Adenine-induced nephropathy

We used an established adenine diet model to induce kidney disease in mice[11,13]. Mice were fed with a standard chow diet supplemented with 0.2% adenine (SAFE, Augy, France) for 3 non-consecutive weeks and then placed back on standard chow diet for the rest of the protocol. Control animals received matched standard chow diet for the entirety of the study.

### Indirect calorimetry

Energy expenditure was assessed using indirect calorimetry (PhenoMaster, TSE Systems). The concentrations of oxygen and carbon dioxide were monitored at the inlet and outlet of the sealed chambers to calculate oxygen consumption and carbon dioxide production. Each chamber was measured at 15 min intervals. Energy expenditure (kcal/h) as a function of oxygen consumption (ml/min), the respiratory exchange ratio (RER) as the ratio of carbon dioxide production on oxygen consumption, food intake (g/day) and water intake (ml/day) were recorded over a 24 h period after 24 h acclimation in the metabolic chambers. Energy expenditure data were expressed per mouse according to international guidelines[26]. Analyses of covariance (ANCOVA) were also carried out to discriminate surgery and diet effects.

### Grip strength

For grip strength evaluation, each mouse was allowed to grab a grid, with forelimbs and hindlimbs, attached to a force transducer (Bioseb) as it was pulled away by the tail horizontally. Three independent trials were averaged for each mouse.

### Glomerular filtration rate (GFR) measurement

Mice were anesthetized with isoflurane and a fluorescence measurement device (MediBeacon) was fixed transdermally on the previously epilated skin of the back of the mice. A first measurement was performed during one minute to quantify the baseline auto-fluorescence. Next, animals were injected retro-orbitally with sinistrin-FITC (7 mg/100 g of weight) and isolated in cages without access to water during 45 min. After that time, the device was removed and results were analyzed with the software MPD Lab (Medibeacon) as previously described[10].

### Blood analyses

Plasma urea and creatinine concentrations were measured using a multi-parametric biochemistry automate Pentra 400 (HORIBA Medical). Plasma levels of IL6, TNFα, CCL2 and IL1β were measured using a multi-parametric next-generation automated ELISA (Ella, Protein Simple).

### Muscle fiber cross-sectional area (CSA)

*Gastrocnemius* muscles were quickly dissected, mounted in 9% Tragacanth gum (Sigma, G1128), frozen in liquid nitrogen-cooled isopentane, and kept at -80 °C. 10 μM cryosections were blocked with mouse on mouse (M.O.MTM) blocking reagent (MKB-2213, Vector Laboratories) before overnight incubation at 4 °C with anti-laminin (L9393, Sigma) primary antibody in DPBS buffer supplemented with 0.5% BSA (A7030, Sigma). The next day, slides were washed in DPBS and stained with anti-rabbit Far Red-Alexa Fluor 647 (711-175-152, Jackson ImmunoResearch) secondary

antibody, in DPBS buffer supplemented with 0.5% BSA, for 1 h at 37 °C. After washing in DPBS, slides were mounted in Fluoromount G medium (FP-483331, Interchim, Montluçon, France). Whole muscle section images were acquired at a 10x magnification with a wide-field fluorescence video-microscope (Video Microscope Cell Observer, ZEISS, Oberkochen, Germany). Mosaics were then stitched (Zen 2.3 lite, Zeiss). For each sample, altered fibers were manually removed. Segmentation of all muscle fibers from a cryosection was performed using Cellpose[39,40]. Mean fiber cross-sectional area was obtained using a self-developed Python script using all muscle fibers from each section.

### Kidney histology

Formalin fixed kidneys were embedded in paraffin, sectioned in 4 μm thick slices (whole cortex) and stained with Masson Trichroma (Dia-Path#010224). Sections were scanned using a Nanozoomer 2.0 RS (Hamamatsu Photonics SARL, Massy, France) and quantification of positive staining was performed using ImageJ.1 Fiji version software (https://imagej.net/downloads). Briefly, after deconvolution, positive staining is specified and pixels that satisfy the color specification are count by an algorithm in all over the scanned slide. Once the algorithm has been confirmed, the settings was saved in a macrofile for subsequent repeated use. Vessel staining was removed from the analysis.

### RNA extraction and real-time qPCR

Total RNA from muscle tissues was isolated using Qiagen RNeasy kit (Qiagen, GmbH Hilden, Germany) following manufacturer's protocol. Total RNA was treated with DNAase (RNase-Free DNase Qiagen) and quantity was determined on a Nanodrop ND-1000 (Thermo Scientific, Rockford, IL, USA). Reverse-transcriptase PCR was performed using the Multiscribe Reverse Transcriptase method (Applied Biosystems, Foster City, CA). Quantitative Real-time PCR (qRT-PCR) was performed in duplicate using the Quant Studio 5 Real-time PCR system (Applied biosystems). All expression data were normalized by the $2^{(-\Delta Ct)}$ method using *TBP* in mice. Primer sequences and Taqman probes (purchased from Thermo Fischer Scientific) are listed in Supplementary Tables 1 and 2.

### Statistics and reproducibility

All biological replicates were derived from distinct animals. Sample size for each outcome is detailed in figure legends. All statistical analyses were performed using GraphPad Prism 10.1.2 for Windows (GraphPad Software Inc., San Diego, CA). Normal distribution and homogeneity of variance of the data were tested using Shapiro-Wilk and F tests, respectively. Student's unpaired *t*-tests and two-way ANOVA followed by Šídák's multiple comparison tests were applied when a significant interaction was found and when appropriate. Alternatively, Mann-Whitney tests were applied when sample size was ≤ 5 biological replicates. For gene expression and plasma cytokines analyses, multiple unpaired *t*-tests with Holm-Šídák's multiple comparisons were applied. All values in figures are presented as mean ± SEM. Statistical significance was set at $P < 0.05$.

### Reporting summary

Further information on research design is available in the Nature Portfolio Reporting Summary linked to this article.

### Data availability

All data generated or analyzed during this study are included in this article and its Supplementary Information files. The Supplementary Information file contains Supplementary Figs. 1-4. Numerical source data behind each graph presented in this study are publicly available in Figshare under the identifier https://doi.org/10.6084/m9.figshare.25315276.

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

## Acknowledgements

This work was supported by Inserm and grants from AFM-Téléthon (n°23801), Société Francophone du Diabète (Allocation de Recherche SFD

2023), European Foundation for the Study of Diabetes EFSD European Research Programme on "New Targets for Type 2 Diabetes" supported by an educational grant from MSD and EFSD/Boehringer Ingelheim European Research Programme on "Multi-System Challenges in Diabetes", Agence Nationale de la Recherche (ANR-21-CE14-0057-01), and Fondation pour la Recherche Médicale (EQU202303016316) (C.M.). B.L. was supported by a Ph.D. fellowship from Fondation pour la Recherche Médicale (FDT202204014748). We are grateful to the personnel of our animal facility care from UMS006-CREFRE, of the We-Met Facility Core, and Genotoul TRI histology and imaging facility core for excellent technical support. We warmly acknowledge Prof. E. Letavernier, and Drs. J. Hadchouel, G. Crambert, and G. Mithieux for critical review of the manuscript and valuable inputs.

## Author contributions

Conceptualization was developed by B.L. and C.M. The investigation was carried out by B.L., M.L., L.F., M.Br., M.Bu., G.F., C.Ma., M.M., L.M., V.B., E.M., N.V., G.T., C.L. and C.M. Data analysis was performed by B.L., M.L., L.F. and C.M. The original draft was written by B.L. and C.M. Review and editing was carried out by all authors. The project was supervised by B.L. and C.M.

## Competing interests

The authors declare no competing interests.
