## [Peer Review File · Communications Biology]

Reviewers' comments:

Reviewer #1 (Remarks to the Author):

The manuscript by Lair et al. investigates the longitudinal changes in body weight, body composition, and energy metabolism in two common mouse models of CKD-related cachexia: the 5/6 nephrectomy (Nx) and adenine-induced nephropathy models. The authors propose that both models may not be suitable for studying CKD-related cachexia because muscle and body weight loss might be triggered by energy deprivation caused by a significant decrease in food intake during the initial weeks of the adenine diet and surgery. As a result, muscle mass and force recover when mice are switched back to a standard diet in the adenine-induced nephropathy model, or once they have recovered from the surgery procedure.

Overall, the paper is well-written and presents highly informative findings relevant to the field. Here is a list of suggestions to bolster the manuscript's conclusions.

Major Points:

- It is critical to discuss whether these models represent cases of acute/transitory renal failure with adaptive compensatory hypertrophy that might be sufficient to recover renal function. Assessing renal histology at early and late time points would provide valuable insights to clarify whether muscle loss is mediated by fasting-induced atrophy or renal failure.

- It is pertinent to discuss the findings from PMID 26669699, where the authors observed no differences in food intake and increased energy expenditure in the Nx model. Providing energy expenditure data regarding Nx will contribute to the discussion.

Minor Points:

- In the methodology section, include details on how food intake is measured in Fig. 2a and 4f to ensure clarity and reproducibility.

- Indicate the temperature at which mice were investigated in the indirect calorimetry experiments to provide necessary experimental details.

-Solagna et al (Reference number 10) has used the adenine diet model and not the Nx.

Addressing these points will enhance the manuscript's quality and strengthen the support for its conclusions.

Reviewer #2 (Remarks to the Author):

The authors performed longitudinal studies to test whether 2 mouse models of CKD (5/6 nephrectomy and adenine nephropathy) faithfully reproduce skeletal muscle abnormalities seen in patients with CKD. Their main finding is that both models induce a transient decrease in caloric intake and a transient weight loss; however, mice quickly regain weight and lean muscle mass. These findings suggest that care must be used when employing these models to make conclusions about the pathogenesis of CKD-related cachexia.

The manuscript is well written and logical. The figures are easy to follow. A strength of the manuscript include detailed longitudinal studies of caloric intake and weight changes in mouse models of CKD. Based on these studies, investigators should consider that CKD models induce short term changes in caloric intake.

However, there are several limitations to the research.

One concern relates to the external validity and significance. It is not clear that the animal models in the manuscript correlate well with published models. For instance, 5/6 nephrectomized mice generally have a higher peri-surgical mortality and develop more significant renal dysfunction. Similarly, the paper cited by the authors (Kim et al AJP 2021) uses a very different adenine model (0.2% adenine for 1 week followed by 0.15% for 7 weeks). In that paper, these mice do develop significant CKD and demonstrate evidence of abnormal bone/mineral metabolism. So, one explanation of their results is that their animal models simply do not develop advanced CKD. Furthermore, the authors determine renal dysfunction using 2 metrics: BUN and creatinine. However, BUN has significant variability and non-standardized creatinine measurements have the same limitations. Real time GFR measurements can help mitigate this.

The authors imply that published studies do not take the transient weight loss into consideration. However, the expectation in all models of CKD (surgical or toxin-related) is that mice will have short-term weight loss with induction of CKD. Most published studies therefore examine the endpoints related to cachexia at delayed time points (6-12 weeks post CKD induction). Importantly, many of the published studies include samples from human skeletal muscle to validate pathogenic pathways of CKD-related cachexia. In contrast to the authors' findings, these studies show persistent changes in muscle mass, atrogenes, and skeletal muscle histology.

Another limitation to the manuscript is that the authors do not include detailed muscle analysis. For instance, there is no assessment of myofibril size, fatty muscle infiltration or fibrosis.

Reviewer comments:

Reviewer #1 (Remarks to the Author):

The manuscript by Lair et al. investigates the longitudinal changes in body weight, body composition, and energy metabolism in two common mouse models of CKD-related cachexia: the 5/6 nephrectomy (Nx) and adenine-induced nephropathy models. The authors propose that both models may not be suitable for studying CKD-related cachexia because muscle and body weight loss might be triggered by energy deprivation caused by a significant decrease in food intake during the initial weeks of the adenine diet and surgery. As a result, muscle mass and force recover when mice are switched back to a standard diet in the adenine-induced nephropathy model, or once they have recovered from the surgery procedure.

Overall, the paper is well-written and presents highly informative findings relevant to the field. Here is a list of suggestions to bolster the manuscript's conclusions.

We thank the reviewer for the positive and constructive assessment of our work. We think we have satisfactorily addressed all concerns and hope the reviewer will be convinced by the revised manuscript.

Major Points:

- It is critical to discuss whether these models represent cases of acute/transitory renal failure with adaptive compensatory hypertrophy that might be sufficient to recover renal function. Assessing renal histology at early and late time points would provide valuable insights to clarify whether muscle loss is mediated by fasting-induced atrophy or renal failure.

This is indeed a critical point. To validate the degree of renal dysfunction and the non-reversible renal phenotype induced by both CKD models, we performed additional experiments to measure GFR and assess kidney histology and fibrosis. These new data are included within the revised manuscript (lines 121-132, 176-186, 247-259, 356-395) and Fig. 4F, Fig. 4L, Fig. 5B-C, Fig. 6A-C. We further conducted a new experiment in which mice exposed to 3 weeks of adenine diet at 12 weeks of age were followed up to 6 months after switching back to standard chow diet. These data are shown in new Fig. 6. There is no doubt that kidney failure is maintained up to 6 months of follow-up in the adenine-diet model as evidenced by the severely reduced GFR and marked renal fibrosis observed at 6 weeks and 6 months. Yet, we fail to observe any muscle wasting in this model (Fig. 6I-J).

- It is pertinent to discuss the findings from PMID 26669699, where the authors observed no differences in food intake and increased energy expenditure in the Nx model. Providing energy expenditure data regarding Nx will contribute to the discussion.

Energy expenditure data are presented in Fig. 2E-F for the Nx model. We are aware of the data from PMID 26669699 from the Spiegelman's lab. Energy expenditure was measured 2-3 weeks' post-surgery in their study, at a time where we here show a significant energy deficit

due to reduced food intake contrary to their findings. Thus, the reported difference in energy expenditure between sham and Nx in their work did not reach statistical significance ($p=0.07$). The discrepancies between our data and their results could stem from differences in mouse age at the time of surgery, surgical procedures, animal care facility sanitary status, chow diet composition and genetic background.

Minor Points:

- In the methodology section, include details on how food intake is measured in Fig. 2a and 4f to ensure clarity and reproducibility.

Details were added in the *Animals* section of the methodology section.

- Indicate the temperature at which mice were investigated in the indirect calorimetry experiments to provide necessary experimental details.

Energy expenditure was measured in metabolic chambers (TSE) at standard housing temperatures (21-22°C).

- Solagna et al (Reference number 10) has used the adenine diet model and not the Nx.

This has been changed for reference Amaya-Garrido, A., et al. Calprotectin is a contributor to and potential therapeutic target for vascular calcification in chronic kidney disease. *Sci Transl Med* 15, eabn5939 (2023).

Addressing these points will enhance the manuscript's quality and strengthen the support for its conclusions.

Reviewer #2 (Remarks to the Author):

The authors performed longitudinal studies to test whether 2 mouse models of CKD (5/6 nephrectomy and adenine nephropathy) faithfully reproduce skeletal muscle abnormalities seen in patients with CKD. Their main finding is that both models induce a transient decrease in caloric intake and a transient weight loss; however, mice quickly regain weight and lean muscle mass. These findings suggest that care must be used when employing these models to make conclusions about the pathogenesis of CKD-related cachexia.

The manuscript is well written and logical. The figures are easy to follow. A strength of the manuscript include detailed longitudinal studies of caloric intake and weight changes in mouse models of CKD. Based on these studies, investigators should consider that CKD models induce short term changes in caloric intake.

However, there are several limitations to the research.

We are grateful to the reviewer for constructive assessment of our work. We think we have satisfactorily addressed all concerns and hope the reviewer will be convinced by the revised manuscript.

One concern relates to the external validity and significance. It is not clear that the animal models in the manuscript correlate well with published models. For instance, 5/6 nephrectomized mice generally have a higher peri-surgical mortality and develop more significant renal dysfunction. Similarly, the paper cited by the authors (Kim et al AJP 2021) uses a very different adenine model (0.2% adenine for 1 week followed by 0.15% for 7 weeks). In that paper, these mice do develop significant CKD and demonstrate evidence of abnormal bone/mineral metabolism. So, one explanation of their results is that their animal models simply do not develop advanced CKD. Furthermore, the authors determine renal dysfunction using 2 metrics: BUN and creatinine. However, BUN has significant variability and non-standardized creatinine measurements have the same limitations. Real time GFR measurements can help mitigate this.

This is indeed a critical point. As also discussed in response to R1 point#1, to validate the degree of renal dysfunction and the non-reversible renal phenotype induced by both CKD models, we performed additional experiments to measure GFR and assess kidney histology and fibrosis. These new data are included within the revised manuscript (lines 121-132, 176-186, 247-259, 356-395) and Fig. 4F, Fig. 4L, Fig. 5B-C, Fig. 6A-C. We further conducted a new experiment in which mice exposed to 3 weeks of adenine diet at 12 weeks of age were followed up to 6 months after switching back to standard chow diet. These data are shown in new Fig. 6. There is no doubt that kidney failure is maintained up to 6 months of follow-up in the adenine-diet model as evidenced by the severely reduced GFR and marked renal fibrosis observed at 6 weeks and 6 months. Yet, we fail to observe any muscle wasting in this model (Fig. 6I-J).

The authors imply that published studies do not take the transient weight loss into consideration. However, the expectation in all models of CKD (surgical or toxin-related) is that mice will have short-term weight loss with induction of CKD. Most published studies therefore examine the endpoints related to cachexia at delayed time points (6-12 weeks post CKD induction). Importantly, many of the published studies include samples from human skeletal muscle to validate pathogenic pathways of CKD-related cachexia. In contrast to the authors' findings, these studies show persistent changes in muscle mass, atrogenes, and skeletal muscle histology.

We have now performed additional experiments to investigate the influence of age (6 week-versus 12 week-old) at the time of surgery on the potential link between CKD and muscle wasting. These data are presented in manuscript (lines 121-132) and new Fig. 4. We have also included a new paragraph in discussion to provide potential explanations of discrepancies between our results and previous reports (lines 249-261).

Another limitation to the manuscript is that the authors do not include detailed muscle analysis. For instance, there is no assessment of myofibril size, fatty muscle infiltration or fibrosis.

We have now performed additional experiments and completed our data with muscle histology to show measures of CSA. Overall, these data confirm the absence of muscle wasting in any

CKD models investigated here. These new data are presented in the revised manuscript (lines 104-111) and in Fig. 3C-D and 3G-H for the Nx model.

REVIEWERS' COMMENTS:

Reviewer #1 (Remarks to the Author):

The authors satisfactorily addressed all of the concerns I raised during the review process

Reviewer #2 (Remarks to the Author):

The authors have answered all questions and the added experiments are sufficient.

I would point out that the renal dysfunction in the adenine model at the later time point was quite mild, other published studies have shown a more severe phenotype.

I would also disagree with the strong statements in the discussion "Collectively, it is likely that no causal link exists between kidney function and muscle mass/force in any other CKD model"

There are numerous variations to these models and it's certainly possible that variations to these models could be more likely to induce skeletal muscle wasting. For instance, adenine 0.15% chow has been used to induce CKD and dietary intake is much higher than with 0.2% chow. I think the main point of the research is quite sound; investigators studying muscle wasting from CKD need to closely assess dietary intake and other measures of muscle wasting. But the authors should acknowledge that other published studies looking at these pathways have used different variations of these models.

Reviewer comments:

Reviewer #1 (Remarks to the Author):

The authors satisfactorily addressed all of the concerns I raised during the review process.

We thank the reviewer for the positive and constructive assessment of our work.

Reviewer #2 (Remarks to the Author):

The authors have answered all questions and the added experiments are sufficient.

I would point out that the renal dysfunction in the adenine model at the later time point was quite mild, other published studies have shown a more severe phenotype.

I would also disagree with the strong statements in the discussion "Collectively, it is likely that no causal link exists between kidney function and muscle mass/force in any other CKD model"

There are numerous variations to these models and it's certainly possible that variations to these models could be more likely to induce skeletal muscle wasting. For instance, adenine 0.15% chow has been used to induce CKD and dietary intake is much higher than with 0.2% chow. I think the main point of the research is quite sound; investigators studying muscle wasting from CKD need to closely assess dietary intake and other measures of muscle wasting. But the authors should acknowledge that other published studies looking at these pathways have used different variations of these models.

We thank the reviewer for the positive and constructive assessment of our work. We understand the remaining concern related to the discrepancy between our study and several previously published studies using these CKD models. We acknowledge that several experimental confounding factors may be at play to explain such discrepancy. As discussed in the manuscript (lines 251-262), this could be due to the surgical procedure, housing conditions, sanitary status of animal care facility and chow diet composition. For instance, adenine supplementation can vary from 0.15 to 0.3% in chow diets according to studies. Yet, feeding with any of these modified diets still produces a major weight loss.

We have tone down our statement lines 262-264.

We observed a ~75% reduction of GFR in mice 25 weeks after removal of the adenine supplementation. This seems to be a quite severe renal dysfunction in line with many other published studies (PMID: 34121452; 37733876; 31461346). We were not able to find any studies with a more severe reduction of GFR. The reduction of GFR induced by the adenine diet model seems also comparable to those induced by 5/6 nephrectomy in our hands and in other published studies (PMID: 34121452). In addition, our longitudinal follow-up in the adenine diet model from 6 weeks up to 6 months after adenine diet removal also reveal that the degree of renal dysfunction remained quite stable in terms of kidney histology and fibrosis.